# Theanine, the Main Amino Acid in Tea, Prevents Stress-Induced Brain Atrophy by Modifying Early Stress Responses

**DOI:** 10.3390/nu12010174

**Published:** 2020-01-08

**Authors:** Keiko Unno, Akira Sumiyoshi, Tomokazu Konishi, Michiko Hayashi, Kyoko Taguchi, Yoshio Muguruma, Koichi Inoue, Kazuaki Iguchi, Hiroi Nonaka, Ryuta Kawashima, Sanae Hasegawa-Ishii, Atsuyoshi Shimada, Yoriyuki Nakamura

**Affiliations:** 1Tea Science Center, University of Shizuoka, 52-1 Yada, Suruga-ku, Shizuoka 422-8526, Japan; gp1728@u-shizuoka-ken.ac.jp (M.H.); gp1719@u-shizuoka-ken.ac.jp (K.T.); yori.naka222@u-shizuoka-ken.ac.jp (Y.N.); 2School of Pharmaceutical Sciences, University of Shizuoka, 52-1 Yada, Suruga-ku, Shizuoka 422-8526, Japan; iguchi@u-shizuoka-ken.ac.jp; 3Institute of Development, Aging and Cancer, Tohoku University, 4-1 Seiryo-machi, Aoba-ku, Sendai 980-8575, Japan; p012024@gmail.com (A.S.); hiroi.nonaka.b1@tohoku.ac.jp (H.N.); ryuta@tohoku.ac.jp (R.K.); 4National Institutes for Quantum and Radiological Science and Technology, 4-9-1, Anagawa, Inage-ku, Chiba 263-8555, Japan; 5Faculty of Bioresource Sciences, Akita Prefectural University, Shimoshinjo Nakano, Akita 010-0195, Japan; konishi@akita-pu.ac.jp; 6College of Pharmaceutical Sciences, Ritsumeikan University, 1-1-1 Nojihigashi, Kusatsu, Shiga 525-8577, Japan; ph0063hv@ed.ritsumei.ac.jp (Y.M.); kinoue@fc.ritsumei.ac.jp (K.I.); 7Faculty of Health Sciences, Kyorin University, 5-4-1 Shimorenjaku, Mitaka, Tokyo 181-8612, Japan; sanae_ishii@ks.kyorin-u.ac.jp (S.H.-I.); ats7@ks.kyorin-u.ac.jp (A.S.)

**Keywords:** brain atrophy, chronic stress, hippocampus, MRI analysis, prefrontal cortex, theanine, SAMP10

## Abstract

Chronic stress can impair the health of human brains. An important strategy that may prevent the accumulation of stress may be the consumption of functional foods. When senescence-accelerated mice prone 10 (SAMP10), a stress-sensitive strain, were loaded with stress using imposed male mouse territoriality, brain volume decreased. However, in mice that ingested theanine (6 mg/kg), the main amino acid in tea leaves, brain atrophy was suppressed, even under stress. On the other hand, brain atrophy was not clearly observed in a mouse strain that aged normally (Slc:ddY). The expression level of the transcription factor *Npas4* (neuronal PAS domain protein 4), which regulates the formation and maintenance of inhibitory synapses in response to excitatory synaptic activity, decreased in the hippocampus and prefrontal cortex of stressed SAMP10 mice, but increased in mice that ingested theanine. Lipocalin 2 (*Lcn2)*, the expression of which increased in response to stress, was significantly high in the hippocampus and prefrontal cortex of stressed SAMP10 mice, but not in mice that ingested theanine. These data suggest that *Npas4* and *Lcn2* are involved in the brain atrophy and stress vulnerability of SAMP10 mice, which are prevented by the consumption of theanine, causing changes in the expression of these genes.

## 1. Introduction

It is well established that stress—especially chronic stress—seriously perturbs physiological and/or psychological homeostasis and affects cognitive function and brain structure, including that of the hippocampus, prefrontal cortex and amygdala [1,2]. For example, in humans, the cumulative exposure to adverse life events is associated with a smaller gray matter volume in the prefrontal and limbic regions which are involved in stress [3]. Chronic restrained stress significantly decreased hippocampal volume and impaired hippocampal neurogenesis in mice [4]. In addition, animal models and human neuroimaging studies have suggested that stress-associated changes are mediated in part by glucocorticoids that are released from the adrenal gland in response to stressors, while circadian glucocorticoid oscillations are disrupted by chronic stress [5,6,7]. Neurogenesis in the hippocampus occurs throughout life in a wide range of animal species and could be associated with hippocampus-dependent learning and memory [8,9,10]. Hippocampal neurogenesis reportedly plays an important role in the regulation of the inhibitory circuitry of the hippocampus [11]. In addition, the maintenance of a balance between inhibitory and excitatory elements in the brain is believed to be important for synaptic plasticity and cognitive function [12,13], and the regulation of inhibitory neuronal activation may be especially important in the hippocampus during chronic stress [14,15,16,17].

We have demonstrated that long-term psychosocial stress using imposed male mouse territoriality, via confrontational housing, accelerates age-related alterations such as cerebral atrophy, oxidative damage, a shortened lifespan, cognitive dysfunction and behavioral depression in the senescence-accelerated mouse prone 10 (SAMP10) [18]. The average survival time of SAMP10 was about 18 months, but under confrontational housing, that was shortened to 14 months. Cognitive dysfunction of SAMP10 was significantly observed at 12 months, but this was already observed at 9 months in stressed mice. On the other hand, in a strain of mice that ages normally (Slc:ddY), a shortened lifespan and cognitive dysfunction were not observed under confrontational housing (unpublished data). These results indicate that aging is accelerated in SAMP10, and stress further accelerates SAMP10 aging. A significant increase in adrenal hypertrophy—a typical marker of the stress response—was similarly observed in ddY mice during confrontational housing [19]. In that case, adrenal hypertrophy developed within 24 h and persisted for at least one week under confrontational housing. However, stress response in SAMP10 mice is considered to continue longer than for ddY mice. Therefore, SAMP10 was used in this experiment as a stress-vulnerable mouse strain, and ddY was used as control.

On the other hand, in both SAMP10 and ddY mice, the intake of theanine—a non-protein amino acid that exists almost exclusively in tea (*Camellia sinensis* L.) leaves—was a potential candidate to suppress psychosocial stress. Although the circadian rhythm of corticosterone was blunted in ddY mice during confrontational housing, it was normalized in mice that ingested theanine (6 mg/kg) even if under the same stressful conditions [18,19]. To examine the reason for cerebral atrophy during confrontational housing, we measured the brain atrophy of SAMP10 and ddY mice using magnetic resonance imaging (MRI). In addition, we examined the mechanism of action of theanine in the brain.

## 2. Materials and Methods

### 2.1. Animals and Preparation of Theanine

Four-week-old male SAMP10 (SAMP10/TaSlc) and ddY (Slc:ddY) mice were purchased from Japan SLC Co. Ltd. (Shizuoka, Japan) and kept in conventional conditions in a temperature- and humidity-controlled room with a 12 h–12 h light–dark cycle (light period, 08.00–20.00 h; temperature, 23 ± 1 °C; relative humidity, 55 ± 5%). Mice were fed a normal diet (CE-2; Clea Co. Ltd., Tokyo, Japan) and water ad libitum. All experimental protocols were approved by the University of Shizuoka Laboratory Animal Care Advisory Committee (approval No. 136068) and were in accordance with the guidelines of the US National Institutes of Health for the care and use of laboratory animals.

L-Theanine (suntheanine; Taiyo Kagaku Co. Ltd., Yokkaichi, Japan) was used at 20 μg/mL and dissolved in normal tap water according to previous methodology [18,19]. Mice consumed theanine solution, which was prepared freshly twice a week, ad libitum. 

### 2.2. Housing Conditions for Stress Experiments

Four-week-old mice were housed in groups of six per cage for five days to habituate them to novel conditions. Mice were then divided into two groups, namely confrontational and group housing, according to a previously described method [18,19]. (Figure 1). In brief, for confrontational housing, a standard polycarbonate cage was divided into two identical subunits by a stainless steel partition. Two mice were housed in the partitioned cage for one month to establish territorial consciousness (single housing). These mice were further divided into two groups that ingested theanine or control water. The partition was then removed to expose the mice to confrontational stress, and the two mice subsequently cohabited in the same cage (confrontational housing). Mice were classified as follows: mice that ingested theanine under confrontational housing were termed CT; mice that ingested control water under confrontational housing, CC; mice that ingested theanine under group housing, GT; and mice that ingested control water under group housing were termed GC. Confrontation periods were 0, 1, 2, 4 and 6 months. The cages were placed in a styrofoam box (width 30 cm; length 40 cm; height 15 cm) in order to avoid visual social contact between cages.

### 2.3. Magnetic Resonance (MR) Sample Preparation

We performed ex vivo MR scanning in this study as it allows for longer scan times, higher resolution scans, and the use of a contrast agent such as a gadolinium complex [20,21,22]. Mouse brain samples were prepared according to the most widely used protocol in the literature [23]. Mice were anaesthetized with isoflurane (N01AB06, Pfizer Inc., Tokyo, Japan) and transcardially perfused with 30 mL of phosphate buffered saline (PBS) containing 2 mM of ProHance (V08CA04, Bracco-Eisai Co. Ltd., Tokyo, Japan), a gadolinium-based MR contrast agent. Subsequently, fixation was performed with 30 mL of 4% paraformaldehyde (PFA, 30525-89-4, Wako, Tokyo, Japan) that also contains 2 mM of ProHance. Bodies, along with the skin, lower jaw, ears, and the cartilaginous nose tip, were removed. The remaining skull structures containing the brain tissue were postfixed in 4% PFA + 2 mM ProHance at 4 °C overnight and then transferred to PBS + 0.01% sodium azide (26628-22-8, Wako, Tokyo, Japan) + 2 mM ProHance at 4 °C in a 15 mL plastic tube. The above procedures were performed at the University of Shizuoka (Shizuoka, Japan) and the ex vivo brain samples were transported to Tohoku University (Sendai, Japan) for MR scanning, which was performed within no longer than 5 months after sampling [24]. Immediately prior to scanning, the ex vivo mouse brains were immersed in liquid fomblin (69991-67-9, Sigma-Aldrich, St. Louis, MO, USA), a perflorocarbon that minimizes susceptibility artifacts at the interface and limits sample dehydration during scanning.

### 2.4. MR Acquisition

All MRI data were acquired using a 7.0 T Bruker PharmaScan 70/16 system (Bruker Biospin, Ettlingen, Germany) with a 23 mm diameter birdcage coil that was designed to serve as both as a transmitter and receiver coil for the mouse brain [25]. The operational software of the MRI scanner was Paravision 5 or 6 (Bruker Biospin, Ettlingen, Germany). Prior to the acquisition of MRI data, global magnetic field shimming was performed inside the core and at the region of interest (ROI) using a point-resolved spectroscopic protocol [26]. The line width (full width at half maximum) at the end of the shimming procedure ranged from 15 to 20 Hz in the ROI. As the T1 tissue contrast between grey and white matter is less pronounced at a high magnetic field strength in rodents compared with humans [27], we used T2-weighted imaging in this study, as in our previous rodent studies [25,28,29]. The T2-weighted images were obtained using a spin-echo 3D-RARE (rapid acquisition with relaxation enhancement) pulse sequence with the following parameters: repetition time = 325 ms, effective echo time = 32 ms, RARE factor = 4, flip angle = 90 degrees, field of view = 12 × 12 × 17.4 mm^3^, matrix size = 200 × 200 × 290, voxel size = 0.06 × 0.06 × 0.06 mm^3^, bandwidth = 60 kHz, fat suppression = on, and number of averages = 10. The total MRI scanning time for each mouse brain was approximately 13 h. The MRI acquisition parameters were set to achieve a reasonable signal-to-noise ratio (SNR) of the mouse brain image [30]. The SNR for each T2-weighted image was 43 ± 9 (mean ± standard deviation), which was measured as the mean image intensity in a single slice of the brain divided by the standard deviation of the intensity in the background outside the brain.

### 2.5. MR Image Preprocessing

Each T2-weighted image was reconstructed using Bruker’s Paravision software, exported in Digital Imaging and COmmunications in Medicine (DICOM) format, and converted to The Neuroimaging Informatics Technology Initiative (NIfTI) format using the “dcm2niix” tool [31]. Each image was visually inspected for any possible artifacts, and a total of 237 mouse brain images (*n* = 122 for SAMP10 mice and *n* = 115 for ddY mice) were used for the analysis. Images were processed by a method used in Pagani et al. [32] in which semi-automated registration-based anatomical labeling for a high-resolution ex vivo mouse brain image can be achieved. Unless otherwise specified, computational steps were carried out using the Advanced Normalization Tools (ANTs) software package (version 2.2.0) [33]. ANTs software, which employs the symmetric diffeomorphic normalization (SyN) model, has been demonstrated to be the most accurate intensity-based normalization method among 14 nonlinear canonical registration algorithms [34]. Images were preprocessed as follows. First, each T2-weighted image was manually rotated and translated such that the origin of the coordinates occupied the midpoint of the anterior commissure to roughly match the standard reference space [35]. Second, each image was roughly skull stripped using Analysis of Functional NeuroImages (AFNI) “3dAutomask” tool [36], which estimates the clipping level of the image, and then the lower-intensity clusters were masked out (clip level fraction = 0.5). Third, the image non-uniformity inside the mask was corrected using the “N4BiasFieldCorrection” tool in ANTs [37] with default parameters. Fourth, to further remove the non-brain tissues, each image was segmented using the “Atropos” tool in ANTs, which employs Markov random field theory [38]. A standard k-means clustering algorithm was used to determine six classes, and the top one or two classes (i.e., higher-intensity clusters) were classified as non-brain tissues and finally removed. Fifth, the resulting skull-stripped image was further bias field corrected inside the mask and then used to create the study-specific template.

### 2.6. Minimum Deformation Template

While several mouse brain templates have been reported in the literature [39,40,41], the mouse strains, magnetic field, and imaging sequence, resolution, and contrast are different among studies and do not perfectly match the current study. The creation of a study-specific minimum deformation template (MDT) has been recommended to provide superior registration accuracy between subjects [42]. First, we computed the MDT at each age range for each mouse strain. The number of mice in each group is summarized in Figure 1. Second, the 10 computed MDTs from each group were used to create a single MDT that was consequently derived from all 237 mouse brains. Third, all subject MDTs were linearly (affine) and nonlinearly (SyN) registered to the mouse ex vivo brain template that was reported in the literature [43] and had 20 brain structure labels. All subject MDTs and structure labels are displayed in Figure 2. MDT computation was performed using the script in ANTs named “antsMultivariateTemplateConstruction2.sh” with the following SyN parameters: gradient step size = 0.1 voxels, update field variance = 3 voxels, and total field variance = 0.5 voxels. These SyN parameters were suggested by a recent mouse ex vivo brain study to account for a balance between registration quality and computation time [44]. Other parameters were as follows: iteration of template creation = 4, maximum iterations for each pairwise registration = 30 × 20 × 10, shrink factors = 4 × 2 × 1 voxels, smoothing factors = 2 × 1 × 0 voxels, similarity metric = cross-correlation, radius in brackets = 4 voxels, N4BiasFieldCorrection = on, and type of transformation model = Greedy SyN.

### 2.7. Measurement of DNA Microarray and Principal Component Analyses

The mice were housed confrontationally for three days after single housing for one month. Group-housed mice were kept in a group of six for one month. Mice were anesthetized with isoflurane and blood was removed from the jugular vein. The hippocampus was removed and frozen immediately. Total RNA was extracted from the hippocampus using an RNeasy Mini Kit (74104, Qiagen, Valencia, CA, USA). Total RNA was processed to synthesize biotinylated cRNA using One-Cycle Target Labeling and Control Reagents (Affymetrix, Santa Clara, CA, USA) and then hybridized to a Total RNA Mouse Gene 1.0 ST Array (Affymetrix), with three biological repeats per group. Raw data were parametrically normalized [45] by using the SuperNORM data service (Skylight Biotech Inc., Akita, Japan). The significance of theanine ingestion was statistically tested by two-way ANOVA [46] at *p* < 0.001.

To compare the effects of theanine ingestion in the control under group or confrontational housing, we performed principal component analysis (PCA) [47] on ANOVA-positive genes [48]. To reduce the effects of individual variability among samples, the axes of PCA were estimated on a matrix of each group’s sample means and applied to all data, which were centered using the sample means of control mice under group housing.

### 2.8. Quantitative Real-Time Reverse Transcription PCR (qRT-PCR)

The mice were housed confrontationally from 0 to 7 months after one month of single housing. Group-housed mice were kept in a group of six for two months. Mice were anesthetized with isoflurane and blood was removed from the jugular vein. The brain was carefully dissected and the hippocampus and prefrontal cortex were immediately frozen. The brain sample was homogenized and total RNA was isolated using a purification kit (NucleoSpin^®^ RNA, 740955, TaKaRa Bio Inc., Shiga, Japan) in accordance with the manufacturer’s protocol. The obtained RNA was converted to cDNA using the PrimeScript^®^ RT Master Mix kit (RR036A, Takara Bio Inc.). Real-time quantitative RT-PCR analysis was performed using the PowerUp™ SYBR™ Green Master Mix (A25742, Applied Biosystems Japan Ltd., Tokyo, Japan) and automated sequence detection systems (StepOne, Applied Biosystems Japan Ltd., Tokyo, Japan). Relative gene expression was measured by previously validated primers for *Npas4* [49] and *Lcn2* [50] genes: Npas4 (F; 5′-AGCATTCCAGGCTCATCTGAA-3′, R; 5′-GGGCGAAGTAAGTCTTGGTAGGATT-3′) and Lcn2 (F; 5′-TACAATGTCACCTCCATCCT GG-3′, R; 5′-TGCACATTGTAGCTCTGTACCT-3′). Since these expressions were significantly changed in the hippocampus of SAMP10 mice by theanine ingestion, the levels were compared among mice that were housed confrontationally from 0 days (only single housing) to 7 months and group-housed for 2 months. cDNA derived from transcripts encoding β-actin was used as the internal control.

### 2.9. Statistical Analyses

Each label volume was computed for each mouse using “ImageMath” and “LabelStats” tools in ANTs. Statistical analysis was performed using one-way ANOVA, and statistical significance was set at *p* < 0.05. Confidence intervals and significance of differences in means were estimated by using Tukey’s honest significant difference method or Fisher’s least significant difference test.

## 3. Results

### 3.1. Effect of Psychosocial Stress in SAMP10 and ddY Mice

The effects of confrontational housing and theanine ingestion on brain volume were examined in SAMP10 and ddY mice, respectively. The body weights of mice housed confrontationally were not different from the group-housed mice. The ingestion of theanine also did not affect body weight. The number of mice for each group was 26–36 (Figure 1). Each image was visually inspected for any possible artifacts, and a total of 237 mouse brain images (*n* = 122 for SAMP10 and *n* = 115 for ddY) were used for the analysis. The MDT and structure labels of all subjects are displayed in Figure 2. Brain volume was compared among the four groups (CC, CT, GC and GT) without distinguishing mouse age. Hippocampal volume was significantly lower in SAMP10 mice of CC than CT and GC (Figure 3a), indicating that atrophy was caused under confrontational housing and was significantly suppressed in mice that ingested theanine. The prevention of atrophy by the ingestion of theanine under confrontational housing was similarly observed in the neocortex of SAMP10 mice (Figure 3b). The time-course of brain atrophy was next examined in mice that were housed confrontationally. In the hippocampus, atrophy was significantly suppressed in aged mice that ingested theanine (6 months of confrontation, Figure 3c). As brain atrophy with aging was not significant in mice housed in a group (Figure 3e), it was clarified that psychosocial stress due to confrontational housing promoted brain atrophy in the hippocampus of SAMP10 mice. The neocortex was significantly smaller one month after starting confrontational housing in SAMP10 mice (Figure 3d). While the lower volume recovered after 2 months of confrontation in CT, the volume gradually decreased with aging under confrontational housing (Figure 3d,f). A similar phenomenon was observed in other brain regions such as the caudate putamen, cerebellum, amygdala, olfactory bulb and brainstem (Appendix A).

On the other hand, the effect of confrontational housing tended to be less in ddY mice than in SAMP10 mice, and the preventive effect of theanine was observed in ddY mice housed in a group (Figure 4a,b). The time-course of brain atrophy showed that brain atrophy was observed after one month of confrontational housing but that brain volume gradually increased and recovered within five months (Figure 4c,d). Almost no atrophy was observed in mice that ingested theanine. However, brain atrophy progressed in aged ddY mice in group housing, but was almost suppressed in mice that ingested theanine (Figure 4e,f).

These results indicate that brain atrophy occurred due to the psychosocial stress caused by confrontational housing not only in SAMP10 but also in ddY mice; however, atrophy progressed with aging in SAMP10 mice but was temporary in ddY mice. After temporary atrophy, an increase in brain volume was observed in ddY mice. Theanine ingestion suppressed brain atrophy with aging in both SAMP10 and ddY mice.

### 3.2. Effect of Psychosocial Stress and Theanine Intake on Gene Expression of the Hippocampus in SAMP10

To examine the effect of psychosocial stress on the brain, early gene expression was compared between mice in group and confrontational housing. Furthermore, the effect of theanine ingestion was examined in SAMP10 mice. The mice were housed confrontationally for three days. DNA microarray data of confrontational and group-housed mice that ingested theanine or water (control), obtained using high-density oligonucleotide microarrays, showed 2277 positively expressed genes based on two-way ANOVA (*p* < 0.0034). Principal component analysis (PCA) was applied to these genes. The PC scores for four groups and their gene expression are shown simultaneously in a biplot (Figure 5). To observe differences caused by stress, control mice housed in a group were used as the reference expression. The difference between group and confrontational housing appears on the PC1 axis, while the effect of theanine ingestion coincided with the PC2 axis. The effect of theanine ingestion on the magnitude of gene expression on the PC2 axis was larger in mice housed confrontationally than in group-housed mice. The top 10 biological processes that were significantly observed on the PC2 axis are shown in Table 1. Many processes, such as transcription and phosphorylation, were negatively regulated by theanine ingestion. On the other hand, oxidation-reduction, apoptosis and lipid metabolism were positively regulated. Several genes that were significantly up or down-regulated following theanine ingestion are summarized in Table 2. Neuronal Per-Arnt-Sim (PAS) domain protein 4, *Npas4*, was the most up-regulated gene. Npas4 is a transcription factor and plays a role in the development of inhibitory synapses [51]. Fatty acid binding protein 7, Fabp7, is a marker of neurogenesis [52]. B cell translation gene 2, *Btg2*, is related to the process of neurogenesis with memory function [53]. On the other hand, the expression level of lipocalin 2 (*Lcn2*), which is up-regulated following psychological stress [54], was down-regulated by theanine ingestion. The function of the most down-regulated gene, melanoma antigen, *Mela*, is unknown. In addition, the expression of *Neat1*, nuclear paraspeckle assembly transcript 1, which inhibits apoptosis [55], was down-regulated.

### 3.3. Effect of Theanine Intake on Levels of Npas4 and Lcn2 in the Brain

Since the expression of *Npas4* increased significantly and that of *Lcn2* decreased significantly in the hippocampus of SAMP10 mice that ingested theanine under confrontational housing on the third day (Table 2), the levels in the hippocampus and prefrontal cortex were compared among mice that were housed confrontationally from 0 days (only single housing) to 7 months and group-housed for 2 months. Although the level of *Npas4* expression tended to be high in mice on day 0 of confrontational housing relative to group housing, the levels decreased in mice housed confrontationally for three days and one month but recovered in mice housed confrontationally for 7 months (Figure 6). On the other hand, in the hippocampus of mice that ingested theanine, the level of *Npas4* expression was significantly higher in mice housed confrontationally for one month relative to day 0 of confrontational housing. The level of *Npas4* expression was not different between SAMP10 and ddY mice housed in a group. Although in ddY mice the level of *Npas4* expression was increased by confrontational housing for three days, the level was lowered after one month. The level of *Npas4* expression in the prefrontal cortex decreased in SAMP10 mice housed confrontationally for three days and recovered in mice housed confrontationally for 7 months. In the prefrontal cortex of SAMP10 mice that ingested theanine, the level of *Npas4* expression was significantly higher in mice housed confrontationally for 7 months than in other mice (Figure 6). The level of *Npas4* in ddY mice increased by confrontational housing for three days and lowered after one month. 

The level of *Lcn2* expression was significantly higher in SAMP10 mice housed confrontationally for three days than in other mice (Figure 7). However, a significant increase was not observed in mice that ingested theanine. On the other hand, the level of Lcn2 expression in the hippocampus of ddY mice slightly increased by confrontational housing for three days, but the levels in ddY mice were one-tenth to one-fifth of those in SAMP10 mice. The level of *Lcn2* expression in the prefrontal cortex showed a similar changed to the expression level in the hippocampus (Figure 7).

## 4. Discussion

SAMP10 mice are susceptible to aging and display characteristic age-related cerebral atrophy [56]. We previously found that cerebral atrophy was accelerated in aged SAMP10 mice that were psychosocially stressed by confrontational housing [18]. We examined whether cerebral atrophy caused by psychosocial stress is a specific phenomenon in SAMP10 mice. In this study, volumetric changes induced by psychosocial stress were observed one month after confrontational housing in SAMP10 mice. Furthermore, subsequent brain atrophy with aging was observed after six months of age. On the other hand, in SAMP10 mice that ingested theanine, a reduction in and recovery from cerebral atrophy were observed. Cerebral atrophy was also observed in ddY mice—a strain that ages normally—but it was temporary within three months of age. Since significant adrenal hypertrophy was observed for at least one week after confrontational housing in ddY mice [19], it is considered that SAMP10 and ddY mice similarly feel psychosocial stress in confrontational housing. However, the stress due to confrontational housing may not last long in ddY mice.

Then, we examined the targets of theanine in the brain at the beginning of confrontational housing to determine the reason for the difference between SAMP10 and ddY mice. The expression of *Npas4* was significantly higher in mice that ingested theanine under confrontational housing than control SAMP10 mice. Npas4 is a neuronal transcription factor, the expression of which is enriched in the limbic system [51]. The detection of Npas4 protein in the soma, neurites and synapses suggests that Npas4 is involved in synaptic plasticity in the brain [57]. Using Npas4 knockout neurons, it has been suggested that Npas4 plays an important role in the structural plasticity of neurons [17]. Increased expression of Npas4 is considered to be important for preventing brain atrophy due to stress. Although the expression of Npas4 was reduced by stress loading in SAMP10, it was increased in mice which ingested theanine. On the other hand, in ddY mice, the expression increased during stress loading even without theanine ingestion. The difference between SAMP10 and ddY mice regarding the expression of Npas4 is considered to contribute to the difference in the degree of brain atrophy.

Npas4 expression is considered as a marker of hippocampus activation [58]. Data of *Npas4* knockout mice suggest that Npas4 plays a major role in the regulation of cognitive and social functions in the brain [59]. Although brain atrophy in Npas4 knockout mice is not mentioned, increased *Npas4* expression during stress appears to be needed to increase stress tolerance. That is, while exposure to acute unavoidable stress induced a long-lasting decrease in *Npas4* expression, resilient rats recovered the level of hippocampal Npas4 better than their vulnerable counterparts [60]. Npas4 regulates the formation and maintenance of inhibitory synapses in response to excitatory synaptic activity [59,61]. Npas4 in both excitatory and inhibitory neurons activates distinct programs of late-response genes that promote inhibition in excitatory neurons but induce excitation in inhibitory neurons [62]. Thus, γ-aminobutyric acid (GABA) release is increased, resulting in the overall lowering of the levels of circuit activity [62,63]. These lines of evidence strongly suggest that Npas4 plays an important role in the development of inhibitory synapses by increasing GABA release and lowering the overall levels of circuit activity. The increased expression of *Npas4* by theanine suggests increased GABA release in stressed mice. In addition, theanine inhibits glutamine uptake from the glutamine receptor, resulting in the inhibition of glutamate release [64]. Since chronic stress causes an imbalance of excitation–inhibition generated by a deficit of inhibitory neurotransmitters on principal glutamatergic neurons [65], theanine intake is thought to suppress excessive excitation by increasing the release of GABA through increased *Npas4* expression.

Npas4 has also been demonstrated to play a role in the neuroprotective response in various animal models of acute neurological injury and limits tissue damage through the modulation of the cell death pathway by directing damaged cells to undergo apoptosis instead of necrosis [51]. Furthermore, GABAergic neurons are particularly susceptible to aging-related alterations that are involved in many aging-induced cognitive impairments and brain disorders [66,67]. Increased *Npas4* expression in the prefrontal cortex by theanine may be involved in the suppression of brain atrophy and cognitive decline in aged SAMP10 mice that ingested theanine. Recent data indicates that the suppression of neural excitation by repressor element-1 silencing transcription factor (REST) regulates aging [68]. These studies suppose that increased expression of *Npas4* by theanine may play an important role in suppressing aging by reducing stress.

On the other hand, *Lcn2*, which is induced by acute stress [54,69], was significantly higher in SAMP10 mice three days after confrontational housing but not in mice that ingested theanine. *Lcn2* is up-regulated in the mouse hippocampus following psychological stress [54]. Since the ingestion of theanine suppresses adrenal hypertrophy under stress [19], the suppression of *Lcn2* could be caused by the suppression of excitation of the hypothalamus–pituitary–adrenal axis. The level of *Lcn2* was still high in SAMP10 mice after one month of confrontational housing, but the level in ddY mice was about 1/12 of that of SAMP10 mice. Lcn2, which is primarily secreted by reactive astrocytes, directly induces neuronal damage and amplifies neurotoxic inflammation under many brain conditions [70]. Since Lcn2 protein increases the sensitivity of neuronal cells to cell death [71], long-lasting *Lcn2* overexpression may be a reason for brain atrophy and stress vulnerability in SAMP10 mice. Lcn2 protein regulates cellular iron concentration [72]. Furthermore, Lcn2 modulates several behavioral responses such as cognitive function, depression, neuronal excitability and anxiety [69]. Mucha et al. [54] hypothesize that iron-free Lcn2 acts as an important regulator of neuronal morphological changes under physical conditions, whereas excess iron-free Lcn2 is harmful to neurons by sequestering intracellular iron and shutting down iron-responsive genes. The suppression of excess Lcn2 is thought to be a therapeutic target for chronic neuroinflammatory and neurodegenerative diseases including Alzheimer’s and Parkinson’s diseases, depression, schizophrenia and autism [70], suggesting that theanine may be important in protecting the brain not only from stress but also from many chronic neuroinflammatory and neurodegenerative diseases. Furthermore, the measurement of cerebrospinal fluid Lcn2 levels might be able to diagnose brain damage due to Alzheimer’s disease, traumatic brain injury and chronic stress [73,74].

In addition, neurogenesis may be increased by the increased expression of *Npas4*, *Fabp7* and *Btg2*, because these genes are involved in neurogenesis [49,52,53,75]. The modulation of these genes involved in the early stress response may be important for the suppression of brain injury due to stress. In particular, *Npas4* and *Lcn2* may play key roles in this context. It is necessary to clarify how theanine modulates *Npas4* and *Lcn2* expression in the near future.

## 5. Conclusions

The brain volume of SAMP10—a stress-sensitive mouse—decreased by stress loading. However, theanine—the main amino acid in tea leaves—suppressed brain atrophy. Theanine was suggested to prevent stress-induced brain atrophy by modifying early stress responses such as Npas4 and Lcn2.

## Figures and Tables

**Figure 1 nutrients-12-00174-f001:**
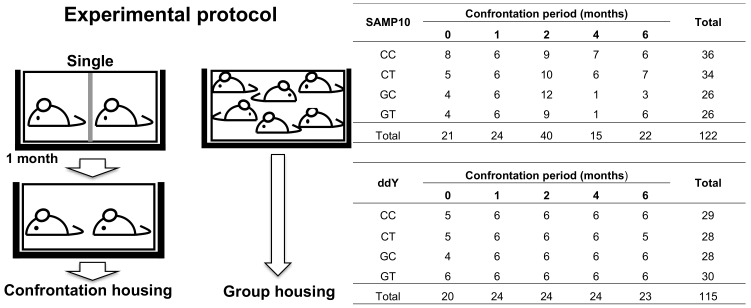
Experimental protocol describing the housing condition. For confrontational housing, a standard polycarbonate cage was divided into two identical subunits by a stainless steel partition. Two mice were housed in the partitioned cage for one month (single housing). These mice were further divided into two groups that ingested theanine or control water. Then, the partition was removed, and subsequently the two mice cohabited the same cage (confrontational housing). Group-housing mice were housed in groups of six. Mice that ingested theanine under confrontational housing, CT; mice that ingested control water under confrontational housing, CC; mice that ingested theanine under group housing, GT; mice that ingested control water under group housing, GC. Confrontation periods were 0, 1, 2, 4 and 6 months. The number of mice for each group is shown.

**Figure 2 nutrients-12-00174-f002:**
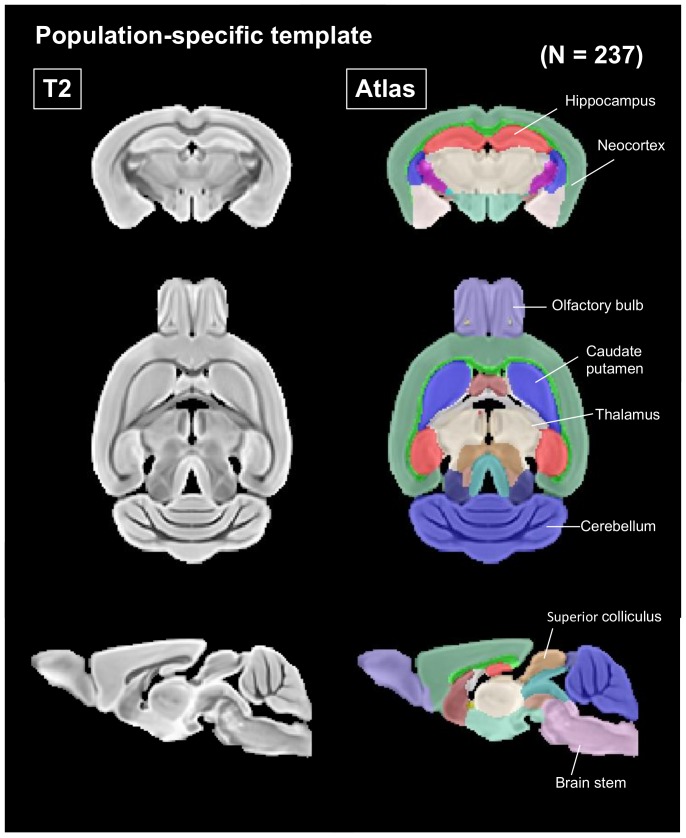
The population-specific minimum deformation template was constructed from the T2-weighted images of all SAMP10 and ddY mice (*n* = 237). The template was nonlinearly registered to the mouse atlas space [43]. Each anatomical structure was used for volume analysis.

**Figure 3 nutrients-12-00174-f003:**
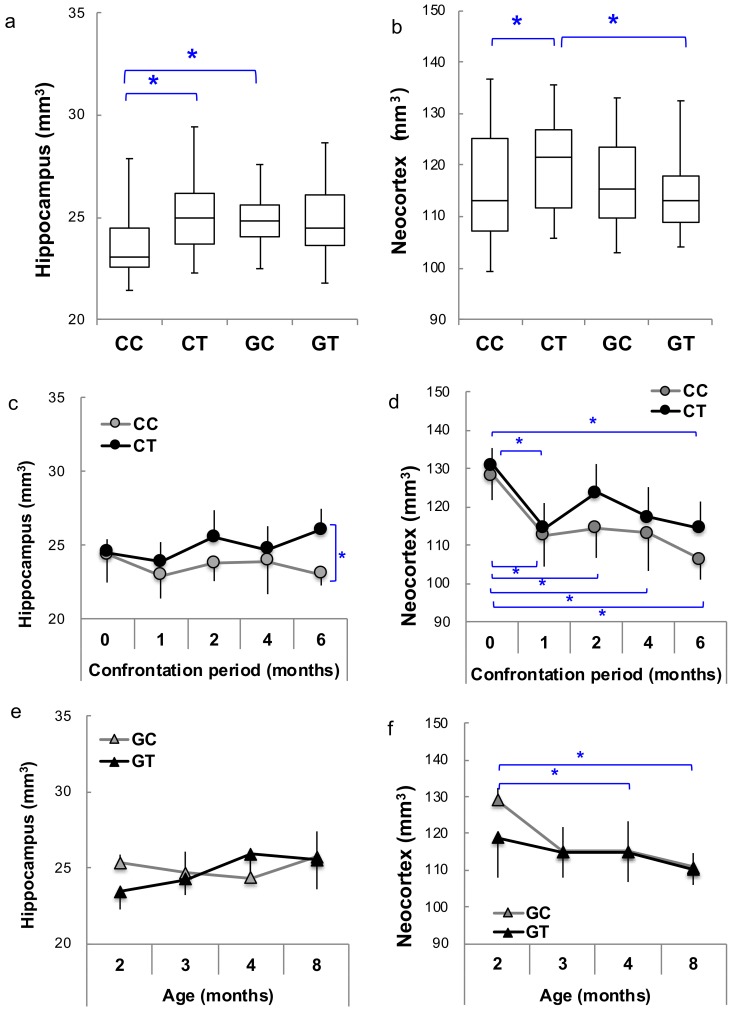
The brain volume of SAMP10 mice. Boxplot of the volumes of brain sections in the hippocampus (**a**) and neocortex (**b**) were compared among the four groups (CC, CT, GC and GT) without distinguishing age (*n* = 122, *; *p* < 0.05). The time-course of hippocampus and neocortex were compared between CC and CT (**c**,**d**) and between GC and GT (**e**,**f**) (*n* = 3–12; *, *p* < 0.05).

**Figure 4 nutrients-12-00174-f004:**
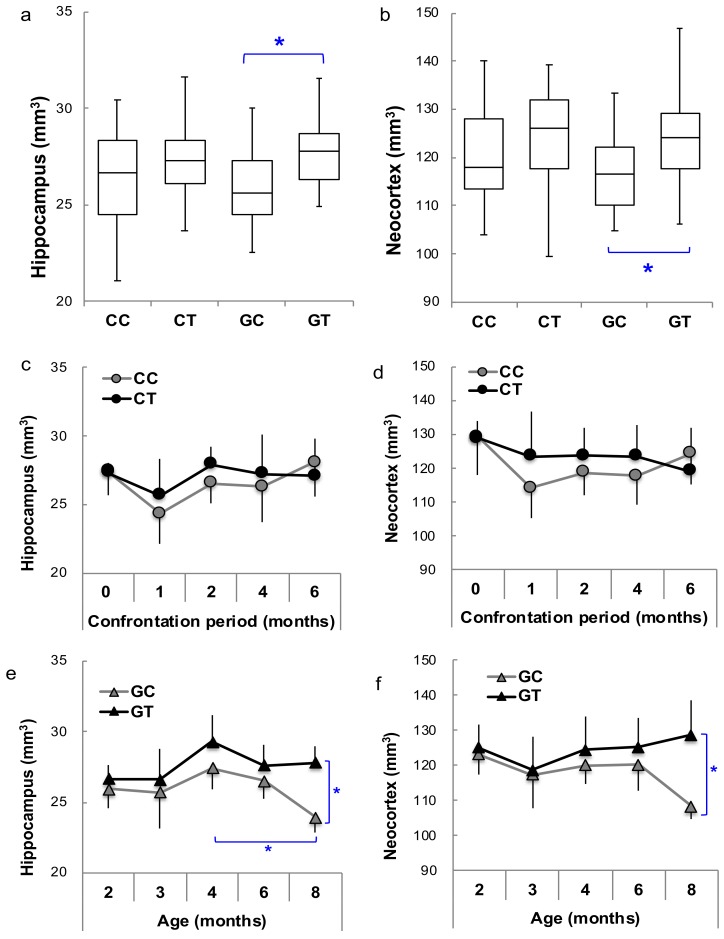
The brain volumes of ddY mice. Boxplot of the volumes of brain sections in the hippocampus (**a**) and neocortex (**b**) were compared among the four groups (CC, CT, GC and GT) without distinguishing age (*n* = 115, *; *p* < 0.05). The time-course of the hippocampus and neocortex were compared between CC and CT (**c**,**d**) and between GC and GT (**e**,**f**) (*n* = 4–6; *, *p* < 0.05).

**Figure 5 nutrients-12-00174-f005:**
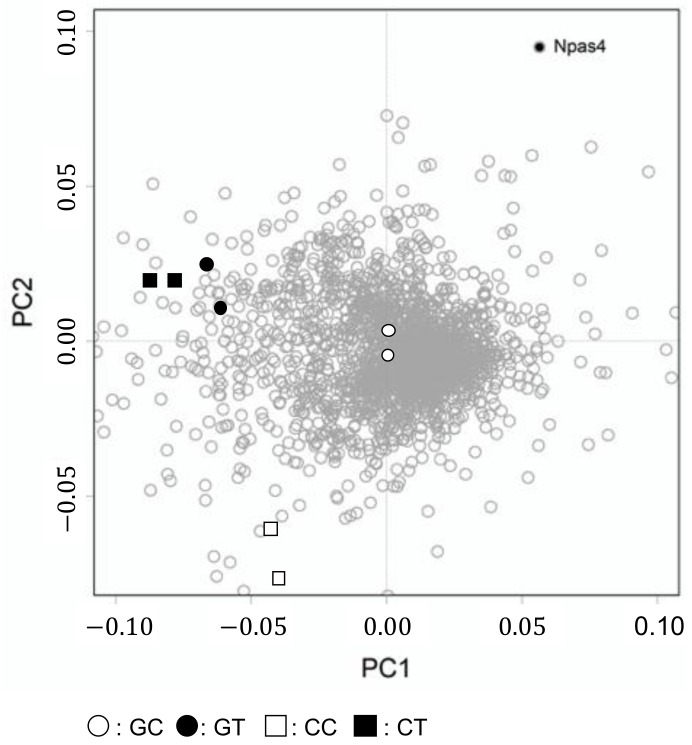
Principal component analysis of gene expression. Hippocampal samples were obtained from mice housed confrontationally for three days. Group-housed mice were kept in a cage for one month. The principal component (PC) ordination of ANOVA-positive genes us based on the transcriptome of hippocampal gene expression in mice of GC (open circle), GT (closed circle), CC (open square) and CT (closed square) groups (*n* = 2 for each group). Each small dot represents the expression of each gene. Neuronal Per-Arnt-Sim (PAS) domain protein 4 (Npas4) is shown as a small black circle.

**Figure 6 nutrients-12-00174-f006:**
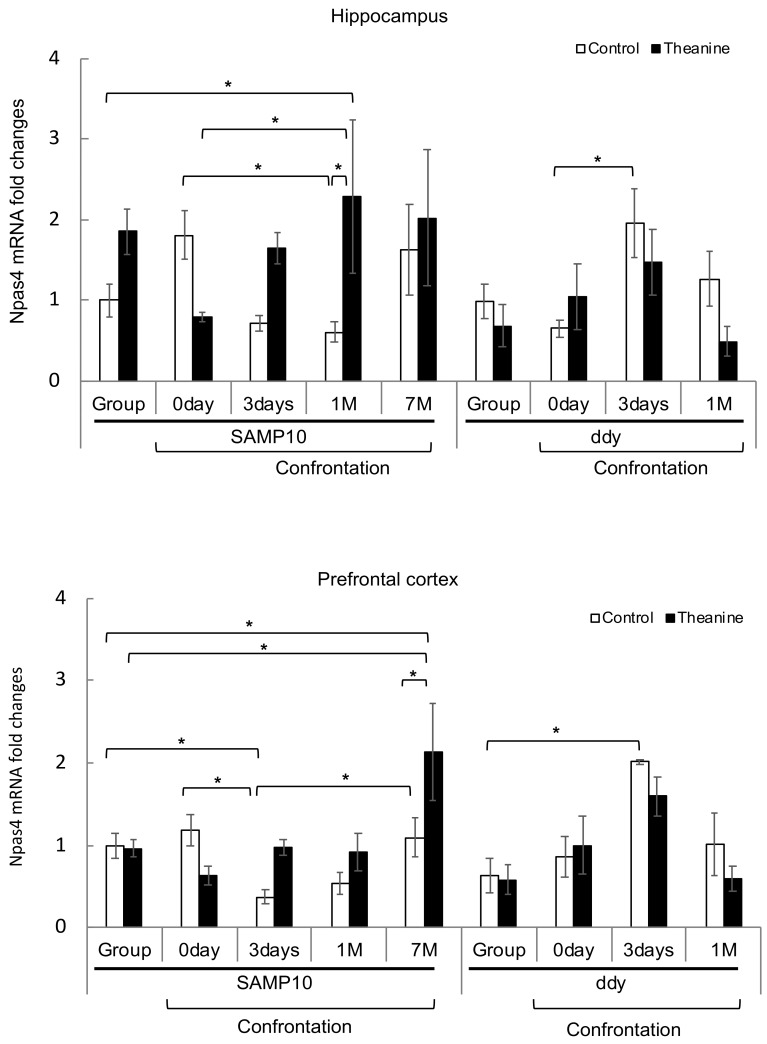
Expression levels of *Npas4* mRNA in the hippocampus and prefrontal cortex of SAMP10 and ddY mice. Mice consumed theanine (20 µg/mL water, closed bar) or normal tap water (control, open bar) ad libitum. After single housing for one month, hippocampal and prefrontal cortex samples were obtained from mice housed confrontationally for 0 days, 3 days, 1 month, and 7 months. Group-housed mice were kept in a cage for two months. Values are expressed as means ± SEM (*n* = 3–6, * *p* < 0.05).

**Figure 7 nutrients-12-00174-f007:**
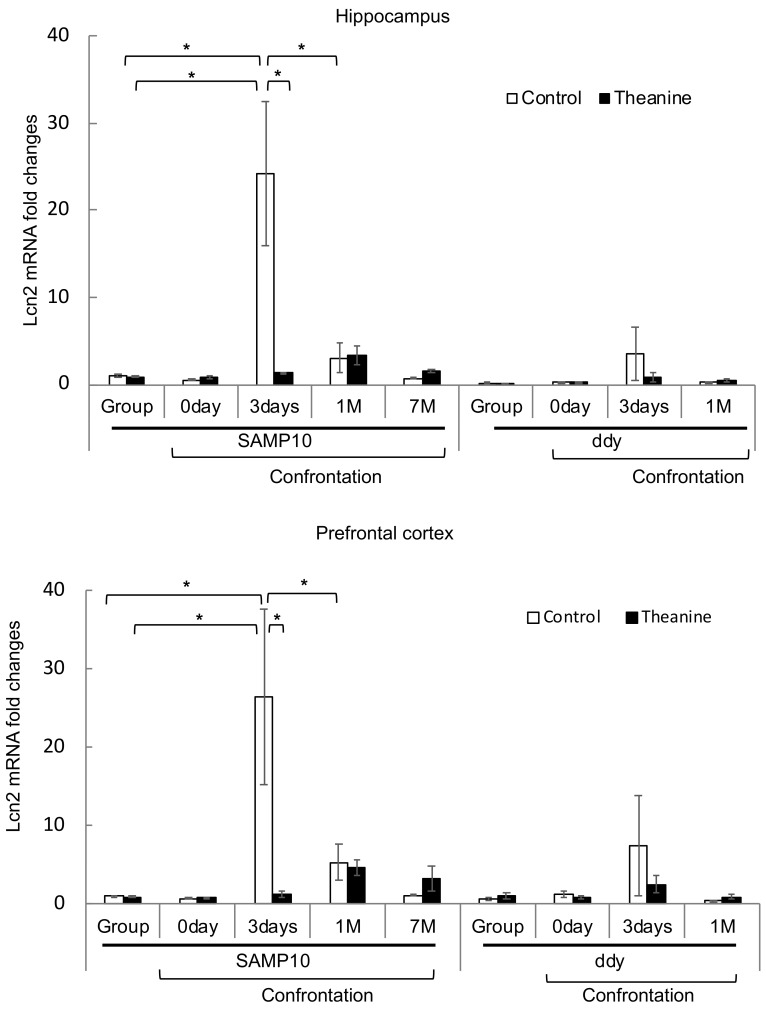
Expression levels of *Lcn2* mRNA in the hippocampus and prefrontal cortex of SAMP10 and ddY mice. Mice consumed theanine (20 µg/mL water, closed bar) or normal tap water (control, open bar) ad libitum. After single housing for one month, hippocampal and prefrontal cortex samples were obtained from mice housed confrontationally for 0 days, 3 days, 1 month, and 7 months. Group-housed mice were kept in a cage for two months. Values are expressed as means ± SEM (*n* = 3–6, * *p* < 0.05).

**Table 1 nutrients-12-00174-t001:** The effect of theanine ingestion on the magnitude of gene expression on the PC2 axis.

PC2	Biological Process	Selected	Total	*p*-Value
Positive	Oxidation-reduction process	40	854	0
	Transport	40	2011	4.22 × 10 ^−7^
	Regulation of transcription, DNA-templated	31	2447	0.0140
	Multicellular organismal development	20	1074	0.0008
	Cell adhesion	16	530	1.32 × 10 ^−5^
	Apoptotic process	16	607	6.54 × 10 ^−5^
	Lipid metabolic process	14	472	5.48 × 10 ^−5^
	Ion transport	13	633	0.0029
	Glucose metabolic process	13	79	3.03 × 10 ^−13^
	Regulation of translation	13	140	3.28 × 10 ^−10^
Negative	Regulation of transcription, DNA-templated	76	2447	0.0002
	Transcription, DNA-templated	70	1983	9.61 × 10 ^−6^
	Signal transduction	69	2582	0.0152
	Transport	69	2011	2.70 × 10 ^−5^
	Metabolic process	45	1595	0.0205
	Multicellular organismal development	42	1074	7.15 × 10 ^−5^
	Cell adhesion	37	530	2.57 × 10 ^−10^
	Phosphorylation	37	737	9.82 × 10 ^−7^
	Protein phosphorylation	37	667	9.30 × 10 ^−8^
	Positive regulation of transcription from RNA Polymerase II promoter	33	899	0.0012

**Table 2 nutrients-12-00174-t002:** Top 10 genes that were significantly up or down-regulated following theanine ingestion.

	Symbol	Full Name	ΔZ	*p*
Up-regulated	Npas4	Neuronal PAS domain protein 4	0.3210	8.32 × 10 ^−21^
	Fabp7	Fatty acid binding protein 7, brain	0.0926	1.25 × 10 ^−31^
	*n*-R5s70	Nuclear encoded rRNA 5S 70	0.2585	0.0028
	Opalin	Oligodendrocytic myelin paranodal and inner loop protein	0.2031	2.67 × 10 ^−24^
	Olfr1474	Olfactory receptor 1474	0.3071	0.0011
	Igkv14–130	Immunoglobulin kappa variable 14–130	0.2886	0.0027
	Vmn2r78	Vomeronasal 2, receptor 78	0.2541	0.0016
	Triml2	Tripartite motif family-like 2	0.2697	0.0032
	Btg2	B cell translocation gene 2, anti-proliferative	0.2027	1.86 × 10 ^−11^
	Igkv18–36	Immunoglobulin kappa chain variable 18–36	0.2167	0.0034
Down-regulated	Mela	Melanoma antigen	−0.6668	6.16 × 10 ^−64^
	Ly6a	Lymphocyte antigen 6 complex, locus A	−0.4346	1.01 × 10 ^−8^
	Lcn2	Lipocalin 2	−0.3125	2.37 × 10 ^−7^
	Prg4	Proteoglycan 4	−0.2707	4.82 × 10 ^−7^
	Neat1	Nuclear paraspeckle assembly transcript 1	−0.3590	1.41 × 10 ^−18^
	C1qb	Complement component 1, q subcomponent, beta polypeptide	−0.2347	1.54 × 10 ^−17^
	Vwf	Von Willebrand factor homolog	−0.2057	7.46 × 10 ^−21^
	C1qc	Complement component 1, q subcomponent, C chain	−0.2948	4.38 × 10 ^−9^
	Hbb-b2	Hemoglobin, beta adult minor chain	−0.2340	1.30 × 10 ^−8^
	Etnppl	Ethanolamine phosphate phospholyase	−0.2038	1.84 × 10 ^−7^

ΔZ = expression level (confrontation theanine–confrontation control).

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
