# Peer review of "Theanine, the Main Amino Acid in Tea, Prevents Stress-Induced Brain Atrophy by Modifying Early Stress Responses"

_nutrients, 2020, doi:10.3390/nu12010174_

Round 1

Reviewer 1 Report

Comments to authors

This review entitled: Theanine, the main amino acid in tea, prevents stress-induced brain

atrophy through modifying early stress responses, examines the therapeutic implications of theanine in reducing stress by evaluating neurological parameters as well as mRNA expression data. This manuscript convincingly indicates that theanine may represent novel therapeutic approaches in blunting stress.

Although the paper is of potential interest, I have several observations:

In the section describing the expression levels of Npas4 and Lcn2 in brain tissues, Have the authors assess the proteins expression levels of these markers. Immunoblots are highly recommended to make a clear and definitive conclusion regarding the modulation levels of these targets. Are these markers analysed in the cerebrospinal fluid of these mice. How authors can explain the modulation levels of these target at this level. Please highlight this point in the discussion. It is well known that stress is highly associated with biological sex. How authors can explain the role of this compound in stressed female mice. How about the implication of this compound in aged male and female mice. Aging is highly associated with stress. Does this compound modulate the levels of cytokines? Androgens? Reactive oxygen species? In plasma or serum. An important point that is highly recommended is to include the catalogue numbers of all the materials and reagents and primers/antibodies that are included in this research. Authors should not use shorthand’s that are hindrance to the reader. Explain abbreviations when they are used for the first time in abstract and text. Authors must use as few abbreviations as possible. A special attention should be given to the English writing of the manuscript; Have the manuscript read and corrected by a colleague proficient in English before submission. The manuscript needs reformulation. Please make a clear and correlated link between the neurological parameters that are assessed and the data of molecular experiments, both in introduction and discussion.

Author Response

This review entitled: Theanine, the main amino acid in tea, prevents stress-induced brain atrophy through modifying early stress responses, examines the therapeutic implications of theanine in reducing stress by evaluating neurological parameters as well as mRNA expression data. This manuscript convincingly indicates that theanine may represent novel therapeutic approaches in blunting stress. Although the paper is of potential interest, I have several observations: 

 In the section describing the expression levels of Npas4 and Lcn2 in brain tissues, Have the authors assess the proteins expression levels of these markers. Immunoblots are highly recommended to make a clear and definitive conclusion regarding the modulation levels of these targets.

Thank you very much for reviewing our manuscript. Since the purpose of this study was to first examine the changes at the transcriptional level caused by theanine intake, changes at the protein level have not yet been confirmed. Certainly, changes in protein expression levels are also important, and we would like to confirm them in the further development of this research.

Are these markers analyzed in the cerebrospinal fluid of these mice?

In this study, we did not measure the levels of these markers in cerebrospinal fluid. However, Lcn2 in cerebrospinal fluid has been reported to be a biomarker for Alzheimer's disease and traumatic brain injury, so Lcn2 might be able to diagnose brain damage due to chronic stress. This was added to the discussion.

How authors can explain the modulation levels of these target at this level? Please highlight this point in the discussion.

Thank you for your valuable suggestion. By focusing on brain atrophy, we sought to explain the importance of Npas4 and Lcn2 modulation levels in discussions.

It is well known that stress is highly associated with biological sex. How authors can explain the role of this compound in stressed female mice?

Since our experimental system was based on the territorial consciousness of male animals, the effect of theanine has not been examined in female mice. However, in clinical studies to date, there has been no difference between women and men in the stress-reducing effect of theanine. Therefore, it is considered that there is almost no sex difference in the action of theanine.

How about the implication of this compound in aged male and female mice?

The effect of aging on the anti-stress effects of theanine has not been clarified. It is necessary to clarify in the future.

Aging is highly associated with stress. Does this compound modulate the levels of cytokines? Androgens? Reactive oxygen species? In plasma or serum.

Although we have not investigated the effects of theanine on cytokines or ROS, it has been reported that theanine (100-200 mg/kg) suppresses the expression of inflammatory cytokines in the brain and reduces damage by reactive oxygen species (Ben P et al 2016, Sumathi T et al 2016). The amount of theanine used in our experiments was about 7 mg/kg, so we do not know how much theanine affected cytokines or ROS in our experiment. However, since Lcn2 is a biomarker of inflammation, it may be related to the suppression of cytokine expression via Lcn2 expression.

There are no reports of effects of theanine on androgens.

An important point that is highly recommended is to include the catalogue numbers of all the materials and reagents and primers/antibodies that are included in this research.

The catalogue numbers of reagents and kits were added. Npas4 and Lcn2 primers referenced previous data. The sequences were added in the methods. Antibody was not used in this experiment.

Authors should not use shorthand’s that are hindrance to the reader. Explain abbreviations when they are used for the first time in abstract and text. Authors must use as few abbreviations as possible.

We explained abbreviations when they are used for the first time in abstract and text, and tried to use as few abbreviations as possible.

A special attention should be given to the English writing of the manuscript; Have the manuscript read and corrected by a colleague proficient in English before submission. The manuscript needs reformulation.

Although our manuscript had been checked by a professional English editing service, the revised version was again checked by MDPI English editing service.

Please make a clear and correlated link between the neurological parameters that are assessed and the data of molecular experiments, both in introduction and discussion.

Many thanks for your valuable review. To make clear, we revised in part of introduction and discussion.

Reviewer 2 Report

In this manuscript, the authors present a novel function of theanine, a non-protein amino acid found abundantly in tea leaves. With some alterations, as outlined below, the manuscript could be improved.

It is presumed that ddy mice are the most appropriate control for SAMP10. But the manuscript would benefit from a more thorough explanation of the relationship between SAMP10 and ddy, and whether either is truly inbred or maintained as an outbred stock. How long do ddy and SAMP10 mice live? When would they be considered “old”? The manuscript references their aging processes, but it would be helpful to know how quickly these mice actually age, especially in comparison to other strains. After all, in B6 mice, 8-months of age is not even middle-age. Has anyone performed similar stress studies on Npas4 knockout mice? If so, that information would be very useful in the Discussion.

Author Response

In this manuscript, the authors present a novel function of theanine, a non-protein amino acid found abundantly in tea leaves. With some alterations, as outlined below, the manuscript could be improved.

Thank you very much for reviewing our manuscript.

It is presumed that ddy mice are the most appropriate control for SAMP10. But the manuscript would benefit from a more thorough explanation of the relationship between SAMP10 and ddy, and whether either is truly inbred or maintained as an outbred stock. How long do ddy and SAMP10 mice live? When would they be considered “old”? The manuscript references their aging processes, but it would be helpful to know how quickly these mice actually age, especially in comparison to other strains. After all, in B6 mice, 8-months of age is not even middle-age.

The strain of ddY is outbred and SAMP10 is inbred.

The longevity of SAMP10 is 1-2 y (average about 18 months), and that of ddY is 2-3 y. When the brain function was examined by a passive avoidance test, a significant decrease was observed at 12 months of age in SAMP10, but that was at 9 months of age in stressed SAMP10 (Unno et al, 2011). Therefore, brain aging in SAMP10 is considered to begin at 8-9 months of age under confrontational housing. On the other hand, ddY mice are considered to be old when they are 20 months or older.

The difference in lifespan between the two strains was added in the introduction.

Has anyone performed similar stress studies on Npas4 knockout mice? If so, that information would be very useful in the Discussion. 

Although there has not been performed similar stress studies, we discussed the relationship between Npas4 expression and brain atrophy by referring Npas4 knockout neuron.